

# A Weekly Arctic Sea-Ice Thickness Data Record from merged CryoSat-2 and SMOS Satellite Data

Robert Ricker[1,2], Stefan Hendricks[1], Lars Kaleschke[3], Xiangshan Tian-Kunze[3], Jennifer King[4], and Christian Haas[1,5]

[1]Alfred Wegener Institute, Helmholtz Centre for Polar and Marine Research, Bremerhaven, Bussestrasse 24, 27570 Bremerhaven, Germany
[2]Univ. Brest, CNRS, IRD, Ifremer, Laboratoire d'Oceanographie Physique et Spatiale (LOPS), IUEM, 29280, Brest, France
[3]Institute of Oceanography, University of Hamburg, Bundesstrasse 53, 20146 Hamburg, Germany
[4]Norwegian Polar Institute, Tromsoe, Norway
[5]York University, Toronto, Canada

*Correspondence to:* Robert Ricker (Robert.Ricker@awi.de)

**Abstract.** Sea-ice thickness on global scale is derived from different satellite sensors using independent retrieval methods. Due to the sensor and orbit characteristics, such satellite retrievals differ in spatial and temporal resolution as well as in the sensitivity to certain sea-ice types and thickness ranges. Satellite altimeters, such as CryoSat-2 (CS2), sense the height of the ice surface above the sea level, which can be converted into sea-ice thickness. However, relative uncertainties associated with this method are large over thin ice regimes. Another retrieval strategy is realized by the evaluation of surface brightness temperature in L-Band microwave frequencies (1.4 GHz) with a thickness-dependent emission model, as measured by the Soil Moisture and Ocean Salinity (SMOS) satellite. While the radiometer based method looses sensitivity for thick sea ice (> 1m), relative uncertainties over thin ice are significantly smaller than for the altimetry-based retrievals. In addition, the SMOS product provides global sea-ice coverage on a daily basis unlike the narrow-swath altimeter data. This study presents the first merged product of complementary weekly Arctic sea-ice thickness data records from the CS2 altimeter and SMOS radiometer. We use two merging approaches: a weighted mean and an optimal interpolation scheme (OI). While the weighted mean leaves gaps between CS2 orbits, OI is used to produce weekly Arctic-wide sea-ice thickness fields. The benefit of the data merging is shown by a comparison with airborne electromagnetic induction sounding measurements. When compared to airborne thickness data in the Barents Sea, the merged products reveal a reduced root mean square deviation of about 0.7 m compared to the CS2 retrieval and therefore demonstrate the capability to enhance the CS2 retrieval in thin ice regimes.

## 1  Introduction

Sea ice is an essential climate variable and affects many climate related processes, such as heat transfer between ocean and atmosphere or ocean circulation, but also marine operations (Meier et al., 2014). For decades, the variability and changes of the ice covered region have been routinely observed by satellite remote sensing of sea-ice extent and area. However, the thickness of sea ice is a crucial parameter for the ice mass balance and is more difficult to observe. Recent satellite altimeter missions such as ICESat or CryoSat-2 (CS2) demonstrated the capability to provide Arctic sea-ice thickness and volume estimates



(Kwok et al., 2009; Laxon et al., 2013). They are used to measure freeboard, the height of the ice or snow surface above the water level, which can be converted into sea-ice thickness assuming hydrostatic equilibrium. CS2 was launched in 2010 and was primarily designed to measure the thickness of thick, perennial ice, since the retrieval method has large uncertainties over thin ice regimes (Wingham et al., 2006). On the other hand, the Soil Moisture and Ocean Salinity (SMOS) mission, launched in 2009, provides brightness temperature observations at microwave frequencies (L-band), which can be exploited for thin ice thickness retrieval (Kaleschke et al., 2012).

Kaleschke et al. (2010) and Kaleschke et al. (2015) drew attention to the complementary nature of the relative uncertainties of CS2 and SMOS ice thickness retrieval methods. Figure 1 shows typical uncertainty maps and the relative uncertainties of CS2 and SMOS monthly mean thickness retrievals from November 2015 and March 2016. While with SMOS relative uncertainties are lowest for thin ice (< 1 m), CS2 relative thickness uncertainties are smaller over thick ice and rise asymptotically towards thinner ice less than 1 m thick. This is due to the fact that CS2 thickness estimates over thin ice rely on the retrieval of small surface elevations slightly higher than sea level while freeboard of thicker ice is much larger (Ricker et al., 2014). Note that the CS2 uncertainties shown here represent statistical uncertainties only. Systematic errors as associated with the usage of a snow climatology or due to variable snow penetration will increase the uncertainty of altimetry based thicknesses (Ricker et al., 2014; Kwok, 2014; Ricker et al., 2015; Armitage and Ridout, 2015).

Besides the different sensitivity to different thickness ranges, both sensor concepts have significantly different swath widths and revisit times, and therefore provide different update rates of sea-ice thickness observations. While CS2 grids are usually based on data composites from one month, SMOS based retrievals provide daily global coverage. Figure 2 compares weekly means of CS2 and SMOS for November 2015 and March 2016. While valid SMOS ice thickness estimates are found mostly in the marginal ice zones, the CS2 ice thickness retrieval covers major parts of the Arctic multiyear ice (MYI). In November, during the freeze-up, SMOS retrievals cover major parts of the Beaufort Sea, Chuckchi Sea, and East Siberian Sea. Towards spring, due to continued ice growth in these regions, the regions with SMOS retrievals retreat southwards, covering major parts of the Bering Sea and the Sea of Okhotsk (Figure 2b). Figure 3 illustrates the number of valid grid cells of the weekly means as shown in Figure 2. The number of grid cells with co-located SMOS and CS2 estimates is less than 2000, while the number of grid cells that contain thickness estimates from CS2 or SMOS only is about 5000, highlighting the complementary coverage of both sensors.

The spatial and interannual variability of sea-ice thickness is driven by dynamics and thermodynamics (Zhang et al., 2000; Kwok and Cunningham, 2016). For an accurate description of the Arctic sea-ice thickness distribution, it is necessary that thick and deformed ice as well as thin ice regimes are represented adequately. Moreover, particularly the formation of new thin ice during the freeze-up characterizes a large area of the ice cover in autumn. In order to detect changes and interannual variabilities in such areas, accurate thin ice thickness estimates with high temporal and spatial resolution are required.

Wang et al. (2016) evaluate six different sea-ice thickness products, including SMOS and CS2, and find that all satellite products as well as the Pan-Arctic Ice-Ocean Modeling and Assimilation System (PIOMAS) overestimate the thickness of thin ice compared to airborne laser altimetry retrievals of NASA's Operation IceBridge. The smallest bias of 0.26 m over thin ice has been found when using the SMOS product.



**Table 1.** Summary of properties of input and output sea-ice thickness products in this study, including CryoSat-2 (CS2), SMOS, the weighted mean (WM) and the OI product (CS2SMOS).

| Product | Temporal res. | Spatial res. | Coverage | Notes and applicability |
|---------|---------------|--------------|----------|-------------------------|
| CS2 (monthly) | 1 month | 25 km | Arctic wide | For studies of multiyear ice and thick first-year ice (> 1 m), high uncertainties for thin ice and in the marginal ice Zone |
| CS2 (weekly) | 1 week | 25 km | Gaps between orbits, sparse at lower latitudes | For studies of multiyear ice and thick first-year ice (> 1 m) where measurements are available, high uncertainties for thin ice (< 1 m) and in the marginal ice Zone |
| SMOS | 1 day | 12.5 km | Arctic wide | For studies of thin ice (< 1 m) |
| WM | 1 week | 25 km | Gaps between CS2 orbits | For studies of multiyear ice and of thin ice, where measurements are available |
| CS2SMOS | 1 week | 25 km | Arctic wide | For Arctic-wide studies on the entire thickness range, uses optimal interpolation |

Considering the depicted complementarity of CS2 and SMOS retrievals and the need for a better representation of thin ice regimes in global-scale sea-ice thickness products, the goal of this study is to provide a merged product of CS2 and SMOS sea-ice thickness retrievals, which has the capability to complete Arctic sea-ice thickness distributions over the entire thickness range with reduced uncertainties. We also aim for a weekly update rate of the merged product. This ensures that we obtain a
5   sufficient coverage of CS2 observations over perennial sea ice, while at the same time, we benefit from the daily update rates of SMOS observations in order to capture ice growth rates in thin ice regions during the freeze-up. We apply two different merging schemes. The first is represented by a weighted mean, based on the individual uncertainties, which only provides estimates at grid cells where weekly observations are available. The second approach uses an optimal interpolation scheme (OI) for Arctic-wide estimates. Table 1 summarizes the input thickness products and the merged products, their temporal and
10  spatial resolution, as well as coverage and applicability depending on study purposes. In order to assess the improvement of the merged products, we use airborne sea-ice thickness data to compare them with co-located data of the merged products.



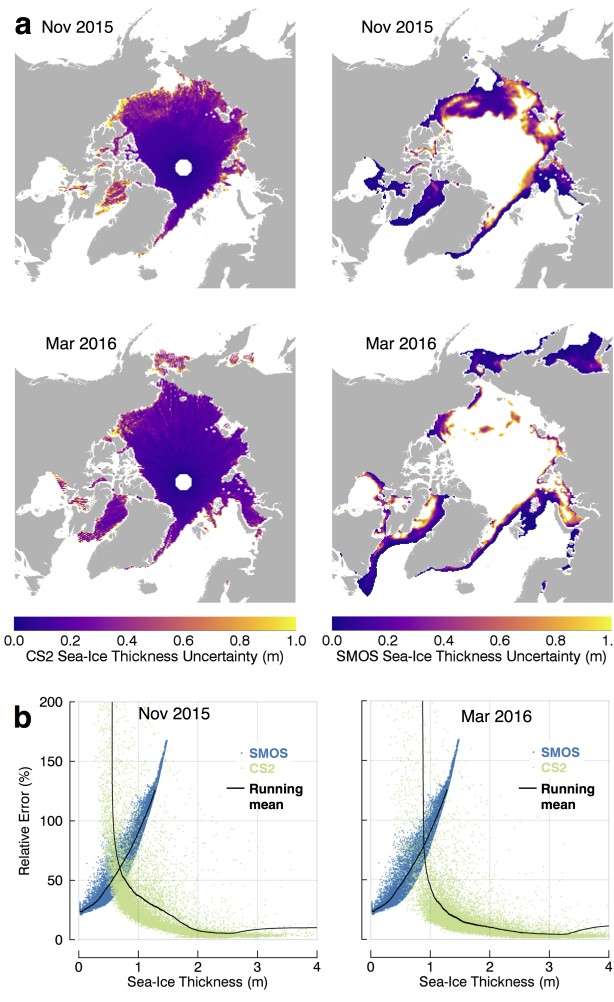

**Figure 1.** a) Typical monthly sea-ice thickness uncertainty maps of the CryoSat-2 and SMOS retrievals from November 2015 and March 2016. The SMOS thickness uncertainty is masked where uncertainty is > 1 m. b) Relative uncertainties from November 2015 and March 2016.





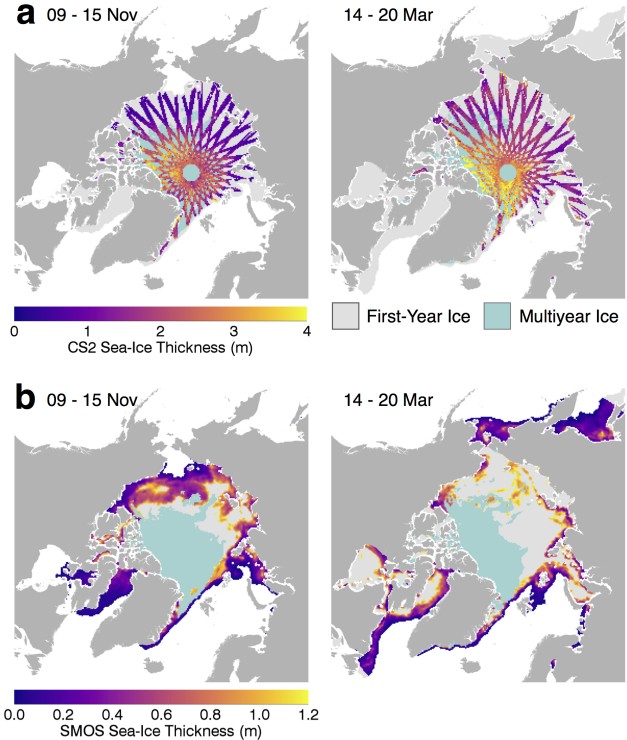

**Figure 2.** Example of weekly input data grids for November 2015 and March 2016. a) Gridded weekly CryoSat-2 retrievals. b) Gridded weekly mean SMOS retrievals derived from daily data. SMOS data are rejected over multiyear ice and when uncertainties are more than 1 m. The background fields indicate first-year and multiyear ice coverage.

## 2 Data and Methods

This section is structured as follows: first, a description of the input data (Section 2.1), then merging weekly CS2 and SMOS data by applying a weighted mean based on the individual uncertainties with the product referred to as WM (Section 2.2), followed by merging weekly CS2 and SMOS data by applying an OI scheme with the product referred to as CS2SMOS (section2.3).

### 2.1 Input Data

We use the AWI CS2 product (processor version 1.2) (Ricker et al., 2014; Hendricks et al., 2016) and the SMOS sea-ice thickness retrieval from the University of Hamburg (processor version 3.1) (Tian-Kunze et al., 2014; Kaleschke et al., 2016) as input ice thickness data. Auxiliary data of ice concentration and ice type were obtained from the Ocean and Sea Ice Satellite Application Facility (OSI SAF).





### 2.1.1 CryoSat-2 Weekly Sea-Ice Thickness Retrieval

In the first step we use CS2 SIRAL Level-1b orbit data files that are provided by ESA. They contain geolocation information and time of the Doppler beam formed radar echoes. SIRAL is operated in two different modes over sea ice. The Synthetic Aperture Radar (SAR) mode covers major parts of the ice covered area, while the Interferometric Mode (SIN) is applied

mostly in coastal areas. Both modes serve for retrieving ice thickness, but must be processed separately.

The radar echoes (waveforms) are processed for each CS2 orbit according to Hendricks et al. (2016) and Ricker et al. (2014). A 50% threshold-first-maximum retracker (Ricker et al., 2014; Helm et al., 2014) is used to obtain ellipsoidal surface elevations ($L$), which are corrected for geophysical perturbations like tides and atmospheric effects (Ricker et al., 2016). Geoid undulations and the mean sea-surface height ($MSS$) are removed by subtracting the Danish Technical University version 2015

(DTU15) $MSS$ height:

$$L_{\mathrm{MSS}} = L - MSS. \tag{1}$$

Ice and water are spatially separated by the pulse peakiness of the CryoSat waveforms. This is based on the fact that radar returns from surfaces that contain open water leads, i.e. openings in the ice pack, appear as specular echoes and can be separated from diffuse echoes that contain reflections from sea ice only. The lead elevations are used to derive the instantaneous sea-

surface height anomaly ($SSHA$) by interpolation. Finally, the SSHA is subtracted from the ice surface elevations to retrieve the freeboard ($Fb$):

$$Fb = L_{\mathrm{MSS}} - SSHA. \tag{2}$$

$Fb$ can be converted into sea-ice thickness ($Z$) by assuming hydrostatic equilibrium (Laxon et al., 2003):

$$Z_{cs2} = Fb \cdot \frac{\rho_{\mathrm{W}}}{\rho_{\mathrm{W}} - \rho_{\mathrm{I}}} + S \cdot \frac{\rho_{\mathrm{S}}}{\rho_{\mathrm{W}} - \rho_{\mathrm{I}}}, \tag{3}$$

where $S$ is the snow depth and $\rho_{\mathrm{S}}$, $\rho_{\mathrm{I}}$, and $\rho_{\mathrm{W}}$ are the densities of snow, sea ice and sea water. S and $\rho_{\mathrm{S}}$ are represented by the modified Warren snow climatology (W99) (Warren et al., 1999). S is reduced by 50 % over FYI to accommodate the recent change towards a seasonal Arctic ice cover (Kurtz and Farrell, 2011). FYI and MYI are separated by adopting the daily OSI SAF ice type product (Eastwood, 2012). We exclude CS2 measurements over Hudson Bay and Baffin Bay as they are not located within the domain of the W99 climatology, which is constrained by in-situ measurements from Soviet drifting stations

and airborne landings from the 1950's to 1990 (Warren et al., 1999). We use ice densities of 916.7 kg/m$^3$ and 882.0 kg/m$^3$ for FYI and MYI (Alexandrov et al., 2010), and 1024 kg/m$^3$ for the sea water density. $Z$ is calculated for each individual CS2 measurement along each orbit. All these retrievals are averaged on a 25 km EASE2 grid (Brodzik et al., 2012) within one calendar week (Figure 2a).

### 2.1.2 SMOS Weekly Sea-Ice Thickness Retrieval

Thin sea-ice thickness has been retrieved from the 1.4 GHz (L-band) brightness temperatures measured by SMOS for the winter seasons (15 Oct -15 Apr) from 2010 to present (Mecklenburg et al., 2016). The retrieval method consists of a thermodynamic





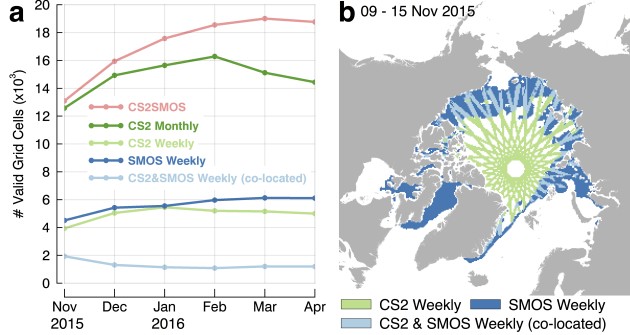

**Figure 3.** a) Numbers of valid 25 km grid cells each month from November 2015 to April 2016. Here, 'valid' grid cells are grid cells that contain a valid thickness estimate. b) Spatial distribution of valid weekly thickness retrievals by CryoSat-2 and SMOS.

sea-ice model and a one-ice-layer radiative transfer model (Tian-Kunze et al., 2014). The resulting plane layer thickness is multiplied by a correction factor assuming a log-normal thickness distribution. The algorithm has been used for the operational production of a SMOS-based sea-ice thickness data set from 2010 on (Tian-Kunze et al., 2014). In this study we use the most up-to-date version (v3.1) of ice thickness data set, which has been produced operationally since October 2016. The v3.1 data

for the previous winter seasons had been reprocessed using the same algorithm.

The v3.1 SMOS ice thickness data are based on v620 L1C brightness temperature data. Brightness temperatures ($TB$) used in the algorithm are the daily mean intensities averaged over incidence angles from 0° to 40°. The intensity is the average of horizontally and vertically polarized brightness temperatures, equal to 0.5 ($TB_h$+$TB_v$). Over sea ice, the intensity is almost independent of incidence angle. By using the whole incidence angle range of 0-40°, we can reduce the brightness temperature

uncertainty to about 0.5 K.

SMOS measurements are strongly influenced by Radio Frequency Interference (RFI), especially in the first two years after its launch. In the previous processor RFI contaminated snapshots have been discarded using a threshold value of 300 K, applied either to $TB_h$ or $TB_v$. The new quality flags given in the v620 L1C data have been implemented to identify the data contaminated by RFI, by sun, or by geometric effects to improve the quality of the radiometric data used for the version 3.1.

To estimate the bulk ice temperature ($T_{ice}$) and bulk ice salinity ($S_{ice}$), which are the important input parameters in the radiation model, we need surface air temperature and sea surface salinity (SSS) data as boundary condition. 2m surface air temperature is extracted from JRA-25 atmospheric reanalysis (Onogi et al., 2007). SSS data used in the retrieval results from an integration of the MIT General Circulation Model (MITgcm) (Marshall et al., 1997), including interannually varying surface forcing. From the daily surface salinity outputs from the model for the years 2002-2009, a weekly climatology was produced

(Tian-Kunze et al., 2014).

Brightness temperatures over sea ice are simulated with the sea-ice radiation model adapted from (Menashi et al., 1993; Kaleschke et al., 2010, 2012). The $TB$ depends on the dielectric properties of the ice layer, which are a function of brine volume (Vant et al., 1978). The brine volume is a function of $S_{ice}$ and $T_{ice}$ (Cox and Weeks, 1983). For a thin ice layer, the ice





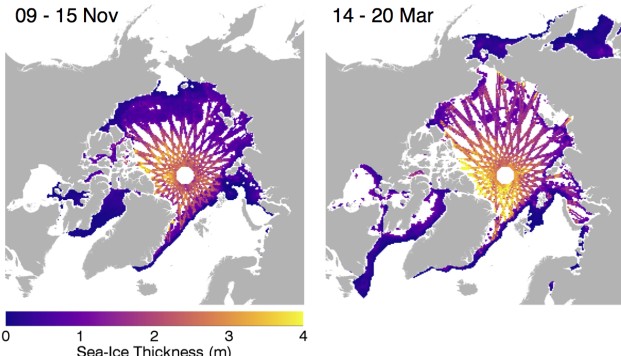

**Figure 4.** Weighted means of CryoSat-2 and SMOS weekly means during the target week, produced from fields shown in Figure 2.

temperature gradient within the ice can be assumed to be linear. The penetration depth of L-band in the sea ice depends on the ice temperature and ice salinity. The retrieval algorithm works only under cold conditions. For the cold and less saline ice, the maximum retrievable ice thickness from SMOS can be up to 1.5 m.

Ice thickness uncertainties are given pixel-wise each day in the data set. There are several factors that cause uncertainties
in the sea-ice thickness retrieval: the uncertainty of the SMOS $TB$, the uncertainties in the ice temperature and ice salinity, and the assumptions made for the radiation and thermodynamic models (Tian-Kunze et al., 2014). In v3.1, we also consider the uncertainty caused by the thickness distribution function. The average ice thickness uncertainty caused by the distribution function is less than 10 cm. A 100% ice coverage assumption made in the retrieval can cause underestimation of ice thickness if the condition is not met.

For the merging, daily SMOS retrievals are averaged weekly and are projected on an EASE2 25 km grid to be co-located with the CS2 retrievals. Here, we only allow SMOS thickness values with a corresponding uncertainty < 1m which corresponds to a maximum theoretical thickness of about 1.1 m. Furthermore we expect strong biases for the SMOS ice thickness in thicker MYI regimes. Therefore we use the OSI SAF ice type product (Eastwood, 2012) to discard any SMOS grid cells that are indicated as MYI. The weekly composites are shown in Figure 2b.

**2.1.3  OSI SAF Ice Concentration and Type**

We use the OSI SAF sea-ice concentration (OSI-401-b) and type (OSI-403-b) products (Eastwood, 2012) in order to identify grid cells that contain $\geq 15$ % sea ice and to classify them as first-year (FYI) or multiyear (MYI) sea ice. The products are delivered daily, projected on a 10 km polar stereographic grid. To combine these data with the CS2 and SMOS thickness grids, we calculate weekly means that are projected on the EASE2 25 km grid (Brodzik et al., 2012) to be co-located with
the thickness retrievals. The original ice type product contains grid cells that are flagged as *ambiguous*. We apply an inverse-distance interpolation to those grid cells to obtain FYI or MYI flags for all ice-covered grid cells, because it is needed for further processing steps.



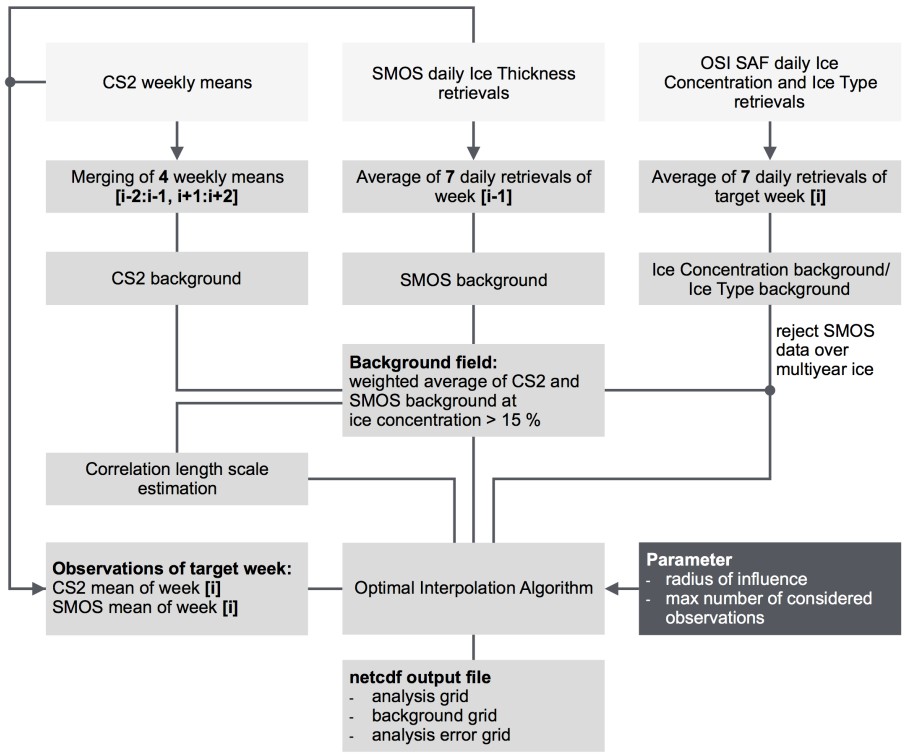

**Figure 5.** Optimal interpolation processing scheme. [i] represents the target week. The cycle is repeated for each week.

## 2.2 Weighted Mean

We compute the weighted mean sea-ice thickness $\overline{Z}$ using weekly CS2 and SMOS ice thickness grids during the target week:

$$\overline{Z} = \frac{Z_{cs2}/\sigma^2_{cs2} + Z_{smos}/\sigma^2_{smos}}{1/\sigma^2_{cs2} + 1/\sigma^2_{smos}}, \tag{4}$$

where $\sigma$ represents the statistical uncertainty of the individual products. Figure 4 shows the weighted means for weeks in November 2015 and March 2016. In contrast to the OI approach, presented in the next section, the weighted mean only provides thickness estimates where observations are available during the target week, leaving data gaps in the CS2 domain. In the following we refer to the weekly weighted mean product as WM.

## 2.3 Optimal Interpolation

To achieve complete spatial coverage, we use an OI scheme similar to Böhme and Send (2005) and McIntosh (1990) that enables the merging of datasets from diverse sources on a predefined, so-called analysis grid. The input data are weighted based on their individual uncertainties and the modeled spatial covariances. OI minimizes the total error of observations and





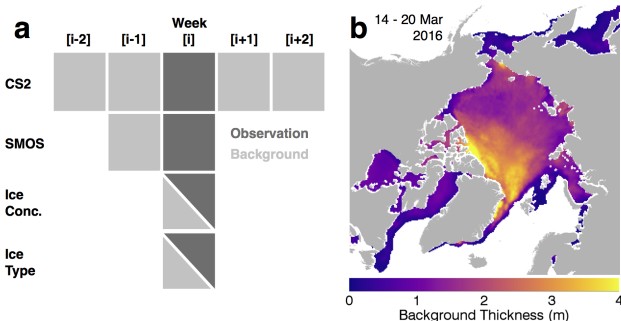

**Figure 6.** a) The scheme illustrates how the background field and the observation field are generated from weekly input grids. [i] represents the target week. b) Interpolated and low-pass filtered background field as it is used for the optimal interpolation.

provides ideal weighting for the observations at each grid cell. In this section we present the processing methods, on which our OI approach is based. Figure 5 shows the processing scheme, which will be described in more detail in the following.

The OI scheme is used to get an objective estimate of values at observed or unobserved locations. The basic equation is:

$$Z_a = Z_b + \mathbf{K}[Z_o - H(Z_b)], \tag{5}$$

where the vector $Z_a$ is the analysis field, i.e. each element represents a grid cell of the merged CS2SMOS ice thickness retrieval to be produced. $Z_b$ is a background field vector and $Z_o$ the vector that contains all SMOS and CS2 observations. Here we use already gridded, weekly mean CS2 and SMOS thickness estimates as observations, as shown in Figure 2 and as described above. Using gridded data as observations reduces their statistical uncertainties and provides equally distributed observations, which improves the performance of the OI. In addition, gridding of raw data reduces the number of available

observations used for the OI, increasing the efficiency of the OI routine. We assume that the observations are static, i.e. remain temporally coherent within a week and do not change due to ice deformation and motion. Therefore, we neglect any temporal correlations. $H$ is an operator that transforms the background field into the observation space. To be more specific, this is realized by an inverse distance interpolation method. $\mathbf{K}$ represents a weight matrix and is derived from error covariances. We aim to retrieve weekly analysis fields, based on calendar weeks from Monday to Sunday. Wet and warm snow or ice prevent

the retrieval of summer sea-ice thickness estimates from CS2 or SMOS. Hence, the CS2SMOS product is limited to the period from October/November to April.

### 2.3.1 The Background Field

The weekly CS2 ice thickness composite possesses large gaps resulting from the limited orbital coverage (Figure 2a). But for the OI approach, an Arctic-wide coverage is required for the background field. Therefore, we use a composite of retrievals

from adjacent weeks, to create a background field with nearly complete coverage for the Central Arctic at a certain target week (Figure 6a). Here we combine data from the two weeks before and after the target week. Therefore, in contrast to CS2 near real-time sea-ice thickness retrievals (Tilling et al., 2016), products can only be released 2 weeks after data acquisition.





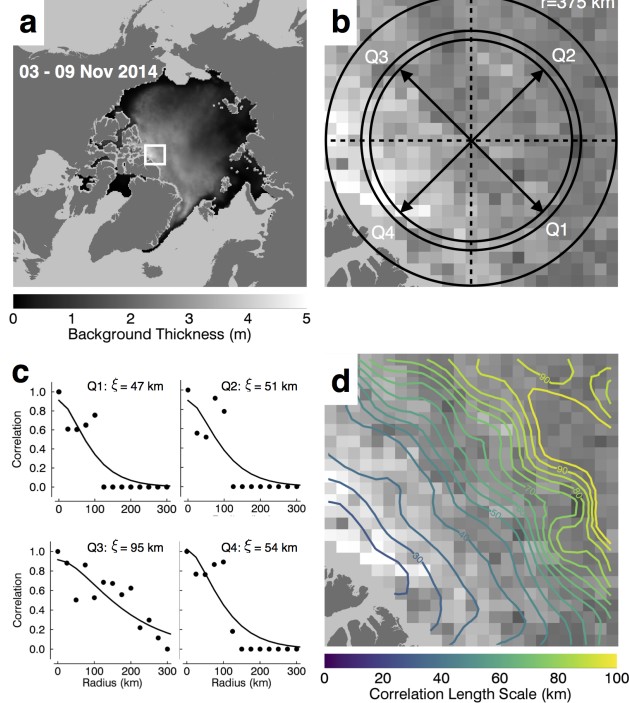

**Figure 7.** Scheme for the estimation of the correlation length scale $\xi$ for a single grid cell for the target week 3-9 November 2014. a) Background field with indicated area of interest (white box). b) Adjacent ice thickness grid cells within a radius of 375 km are binned into annuli of distance and 4 quadrants. (c) Binned thickness estimates are used to calculate the structure function of each quadrant. $\xi$ is estimated by fitting an exponential function. d) Contour map of estimated correlation length scales for the considered area.

In order to ensure independence between observations and background field, CS2 data from the target week are not included in the background field. For the same reason, we use a SMOS weekly mean from the previous week. The initial background field is computed by a weighted mean using Eq. (4). Gaps in the weighted average are interpolated by using a nearest neighbor scheme. In order to reduce noise, the background field is low-pass filtered with a smoothing radius of 25 km, before it is applied

5 in the OI algorithm (Figure 6b).

  Since we use CS2 and SMOS retrievals for the background field beyond the target week and because the SMOS composite contains artifacts in coastal regions, we additionally use a weekly mean of the daily OSI SAF ice concentration product to determine the ice coverage during the target week. Here, we apply a threshold of 15 % and only grid cells that exceed this value will be considered as ice covered, which corresponds to the ice extent products provided by OSI SAF and the National

10 Snow and Ice Data Center (NSIDC).





### 2.3.2  Correlation Length Scale Estimation

The correlation length scale $\xi$ controls the impact of a data point on the analysis grid point depending on their distance. Considering the grid resolution of 25 km, correlation length here is used in the sense of large-scale thickness gradients. For example, the correlation length scale estimate is large in the center of a certain ice type regime with similar ice thickness (i.e. level FYI). On the other hand, we expect a low $\xi$ value at locations with strong thickness gradients, where distant observations are not representative for local conditions. Figure 7 illustrates the estimation of $\xi$ for a certain grid cell $\mathbf{Z_0}$ in the Lincoln Sea during a week in November. In order to estimate $\xi$, we consider the unfiltered background field $\mathbf{Z_b}$ (Figure 7a) and define a structure function $\epsilon^2$. The structure function can be used to assess the change of ice thickness with distance and is related to the normalized auto correlation function $R(d, Q)$ as follows (Böhme and Send, 2005):

$$\epsilon^2(d, Q) = \overline{(Z'_0 - Z'_{Q,d})^2} = 2\overline{\sigma_{Z'}^2} - 2\overline{\sigma_{Z'}^2}R(d, Q),$$

$$R(d, Q) = 1 - \frac{\epsilon^2(d, Q)}{2\overline{\sigma_{Z'}^2}}. \tag{6}$$

Quadrants Q are defined to accommodate the anisotropy of the spatial ice thickness distribution (Figure 7b). $\epsilon^2(d, Q)$ represents the square differences between ice thickness of the grid cell and the ice thickness of the grid cells of binned 25 km distances d in a quadrant Q. $Z'_{Q,d}$ is the background thickness, binned according to $d$ and Q. Figure 7b reveals the annuli of distance and the 4 Quadrants. $\overline{\sigma_{Z'}^2}$ are the corresponding mean variances of a certain quadrant. With Eq. (6) we then obtain the auto correlation function $R(d, Q)$, which is computed up to radius of 750 km (30 bins). In the next step, we fit a function of the form:

$$C(d, \xi) = \left(1 + \frac{d}{\xi}\right) \exp\left(\frac{-d}{\xi}\right) \tag{7}$$

to $R(d, Q)$, using a least squares scheme, and obtain an estimate for $\xi$. Figure 7c shows the calculated auto correlation function $R(d, Q)$ and the functional fit (Eq. (7)). A stronger decay of $R(d, Q)$ occurs with rising deviation between $Z_0$ and the thickness at a certain distance in a certain quadrant. $R(d, Q)$ can also become negative if $\epsilon^2(d, Q)/2\overline{\sigma_{Z'}^2}$ becomes $>1$. In order to improve the fitting performance, we set $R(d, Q) = 0$ if $R(d, Q)$ becomes $< 0$. Furthermore, $\xi$ is rejected if the computation fails. Finally, we average the $\xi$ values from the 4 quadrants, as we do not use anisotropic weighting in the OI. In order to remove outliers and noise, the derived $\xi$ grid is low-pass filtered with a smoothing radius of 25 km. Grid cells with failed computation are interpolated by a nearest neighbor scheme afterwards. Figure 7d shows the spatial correlation length scales $\xi$ for 3-9 November 2014. It highlights the sensitivity to changing thickness gradients as $\xi$ decreases towards the coast of the Canadian Archipelago, where higher sea-ice thickness gradients likely occur due to increased deformation.





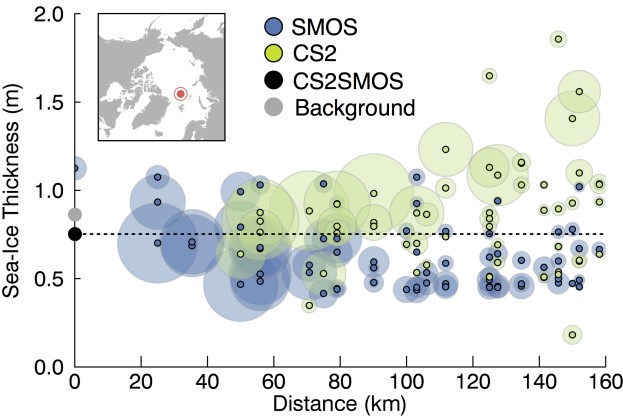

**Figure 8.** Example for CS2 and SMOS sea-ice thickness observations and their weighting to compose the CS2SMOS thickness estimate based on optimal interpolation at a grid cell in the Central Arctic first-year ice in November 2016. The x-axis represents the distance of observations from the analysis grid cell. Normalized K weights are represented by the area of the circles.

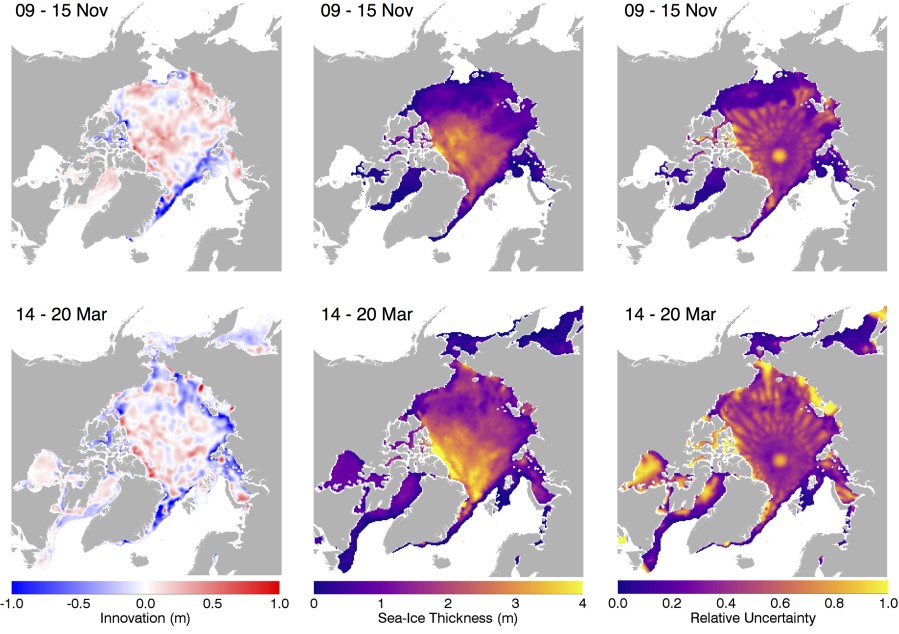

**Figure 9.** Optimal interpolation output grids for weeks in November 2015 and March 2016: The innovation field (left column) shows the difference between background field and the CS2SMOS ice thickness (center column). The right column shows the relative uncertainty associated with the optimal interpolation.



### 2.3.3 Retrieving the Analysis Grid

The weight matrix $\mathbf{K}$, which is needed for the computation of $\boldsymbol{Z_a}$, is retrieved by the background error covariance matrix $\mathbf{B}$ in the observation space, multiplied by the inverted total error covariance matrix:

$$\mathbf{K} = \mathbf{BH}^T(\mathbf{R} + \mathbf{HBH}^T)^{-1}, \tag{8}$$

where $\mathbf{R}$ is the error covariance matrix of the observations. In order to reduce computation expense we assume the following:

1. We neglect correlations of observation errors which means that $\mathbf{R}$ is a matrix with non-zero elements only on the diagonal. These variances are represented by the respective SMOS and CS2 product uncertainties.

2. We assume that the influence of observations that are located far away from the analysis grid point can be neglected. Therefore, instead of computing the entire covariance matrix, we only consider observations within a radius of influence. This radius is set to 250 km to gather just enough observations in regions with large gaps, for example over MYI between two CS2 orbits where valid SMOS observations are not available.

3. To further reduce computation expense we limit the number of matched observations to 120, meaning that in the case of more matches, only the 120 closest observations are considered.

4. We generally assume that all observations are unbiased.

$\mathbf{BH^T}$ and $\mathbf{HBH^T}$ are estimated using the correlation function in Eq. (7):

$$\mathbf{BH^T} = \left(1 + \frac{d(x_{o_i}, x_{a_i})}{\xi}\right) \exp\left(\frac{-d(x_{o_i}, x_{a_i})}{\xi}\right),$$
$$\mathbf{HBH^T} = \left(1 + \frac{d(x_{o_i}, x_{o_j})}{\xi}\right) \exp\left(\frac{-d(x_{o_i}, x_{o_j})}{\xi}\right), \tag{9}$$

with the Euclidian distance function:

$$d(x, y) = \|x - y\| \tag{10}$$

Here, $x_{o_i}$ and $x_{a_i}$ represent the locations of the observations and the analysis grid points. As a consequence of Eq. (9), the impact of a data point decreases with increasing distance.

Computing $\mathbf{BH^T}$ and $\mathbf{HBH^T}$ allows the computation of the $\mathbf{K}$ weights. Thus, we retrieve the second part of Eq. (5), which is called *innovation*, the difference between the observation field and the background field. This iterative procedure is accomplished for each analysis grid cell, leading to the complete analysis grid $\boldsymbol{Z_a}$. The corresponding analysis error covariances are derived by:

$$\sigma^2_{Z_a} = (\mathbf{I} - \mathbf{KH})\mathbf{B}, \tag{11}$$

where $\mathbf{I}$ is the identity matrix. Since we consider variances exclusively, we only calculate the diagonal elements of $\sigma^2_{Z_a}$. Figure 8 illustrates how the analysis thickness is derived at a certain analysis grid point, considering distant grid cells with ice thickness



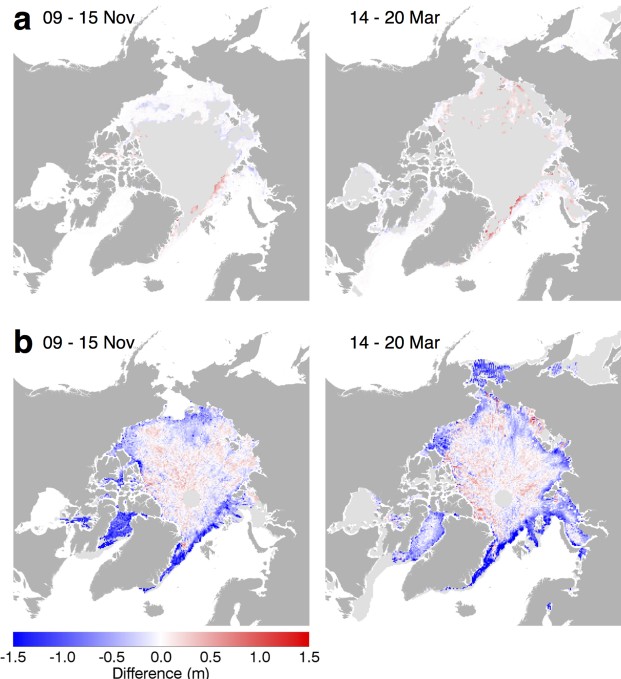

**Figure 10.** a) Difference between CS2SMOS and weekly SMOS retrieval for weeks in November 2015 and March 2016. b) Difference between CS2SMOS thickness for weeks in November 2015 and March 2016, and the corresponding monthly CryoSat-2 thickness retrieval

estimates of CS2 and SMOS. The K weights decrease with increasing distance to the analysis grid point as a consequence of Eq. (9). In addition, the individual uncertainties affect the weighting according to Eq. (8). The considered grid cell is located at the boundary between the CS2 and SMOS domain. In the following, we use *domain* as the regions where CS2 or SMOS data predominate. SMOS ice thicknesses of about 1 m reveal higher uncertainties than corresponding CS2 estimates (Figure 1) and hence the K weights of CS2 estimates exceed the SMOS weights for higher ice thicknesses. Figure 9 shows the innovation field, the merged CS2SMOS product and the analysis error field, which is the square root of the error variance (Eq. (11)), for weeks in November 2015 and March 2016. The analysis error is a relative quantity with values between 0 and 1. It increases where the weekly CS2 retrieval leaves gaps and where valid SMOS observations are not available, for example at the North Pole or over MYI. In this case the analysis depends on the accuracy of the background field, leading to increased uncertainties.

## 3 Evaluation of the Optimal Interpolation

In this section, we aim to evaluate the CS2SMOS product derived from the OI scheme by a comparison with the individual satellite products. In addition, we carry out a cross validation experiment by omission of random data to test the OI method.





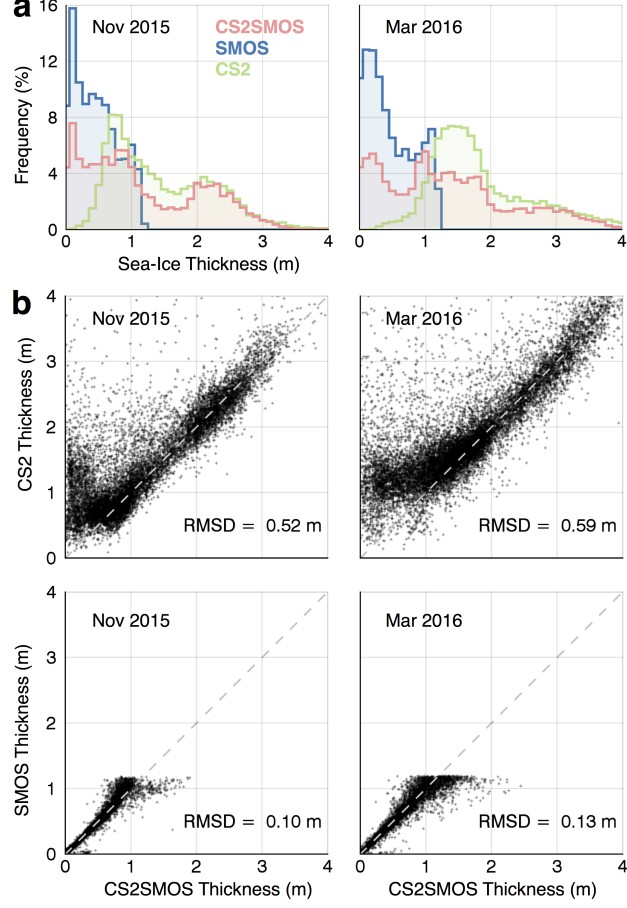

**Figure 11.** a) Sea-ice thickness distributions of CryoSat-2, SMOS, and CS2SMOS retrievals for November 2015 and March 2016. CS2SMOS is represented by one week in the middle of a month, while the CryoSat-2 and SMOS retrievals are monthly means. b) Scatter diagrams illustrating the ice thickness differences between CS2SMOS and the individual satellite retrievals of CS2 and SMOS, for November 2015 and March 2016.

## 3.1 Comparison with Input Products

Figure 10 illustrates the differences between CS2SMOS and the CS2 and SMOS retrievals from November 2015 to April 2016. The difference between CS2SMOS and SMOS weekly grids is shown in Figure 10a, limited to grid cells with SMOS observations in the target week. Positive anomalies of up to 1 m occur mostly in the transition zone between the SMOS and the CS2 domain where the thick ice in the CS2 retrieval leads to an increase of ice thickness in these grid cells with respect to the SMOS data (Figure 10a). However, the general pattern remains the same during the season. Subtracting the CS2 monthly mean sea-ice thickness from the CS2SMOS product, represented by one week within each month, reveals substantial scattering between -1 m and 1 m within the CS2 domain (Figure 10b). This is mainly caused by the fact that the monthly retrieval is





**Table 2.** Arctic-wide mean and standard deviation (sdev) of the merged product (CS2SMOS), the individual CryoSat-2 (CS2) and Soil Moisture and Ocean Salinity (SMOS) retrievals for the winter season 2015/16.

| Mean (m) | Nov | Dec | Jan | Feb | Mar | Apr |
|---|---|---|---|---|---|---|
| CS2SMOS | 1.16 | 1.19 | 1.22 | 1.29 | 1.36 | 1.34 |
| CS2 | 1.46 | 1.53 | 1.65 | 1.66 | 1.83 | 1.90 |
| SMOS | 0.45 | 0.58 | 0.51 | 0.49 | 0.48 | 0.47 |
| Sdev (m) | | | | | | |
| CS2SMOS | 0.88 | 0.8 | 0.81 | 0.92 | 0.96 | 0.99 |
| CS2 | 0.76 | 0.76 | 0.72 | 0.73 | 0.75 | 0.78 |
| SMOS | 0.33 | 0.36 | 0.38 | 0.37 | 0.36 | 0.38 |

compared with the weekly product. During the different time spans, the regional sea-ice thickness distribution is subject to ice drift, convergence and divergence, as well as thermodynamic ice growth. In addition, the OI algorithm evokes a low pass filtering of the spatial thickness distribution due to the impact of distant grid cells, reducing the noise compared to the original

5 CS2 product. Within the SMOS domain we find consistently negative anomalies, indicating a reduction of the CS2 ice thickness representation due to the impact of the coincident SMOS retrieval.

Figure 11a shows ice thickness distributions of monthly means of CS2 and weekly SMOS and CS2SMOS ice thickness retrievals for November 2015 and March 2016, illustrating the different thickness ranges of CS2 and SMOS retrievals. Table 2 presents the corresponding statistics for the entire winter season including the mean and the standard deviation of each month or week respectively. The CS2 retrieval lacks sensitivity for thin ice (< 0.5 m) over the entire season. The gap in this thickness range can be closed by the SMOS retrieval. While the mean thickness of the CS2 retrieval consistently grows from 1.46 m in November to 1.90 m in April, the SMOS thickness mean remains at about 0.5 m after an increase from November to December. Due to the increasing uncertainties of the SMOS product towards thick ice, the frequency steeply drops at about 1 m

for each month. Therefore, the SMOS mean thickness is mostly affected by the boundary condition at about 1 m in conjunction with thermodynamic ice growth and the newly formed ice (< 0.1 m). The thickness distributions show the capability of the CS2SMOS product to combine the complementary ice thickness ranges. As a consequence, the standard deviation of the merged product ranges between 0.8 m (December) and 0.99 m (April), and therefore exceed the standard deviations of the individual products that reach maximum values of 0.78 (CS2) and 0.38 (SMOS) in April. The scatter diagrams in Figure 11b

illustrate the thickness differences between CS2SMOS and the two individual products, with respect to the maps shown in Figure 10. Using the SMOS data reduces the thickness in the CS2SMOS product below 1m compared to the CS2 retrieval. The comparison between CS2SMOS and SMOS shows increasing scattering with rising thickness. As shown in Figure 10, this originates from the transition zone between the CS2 and SMOS domain.



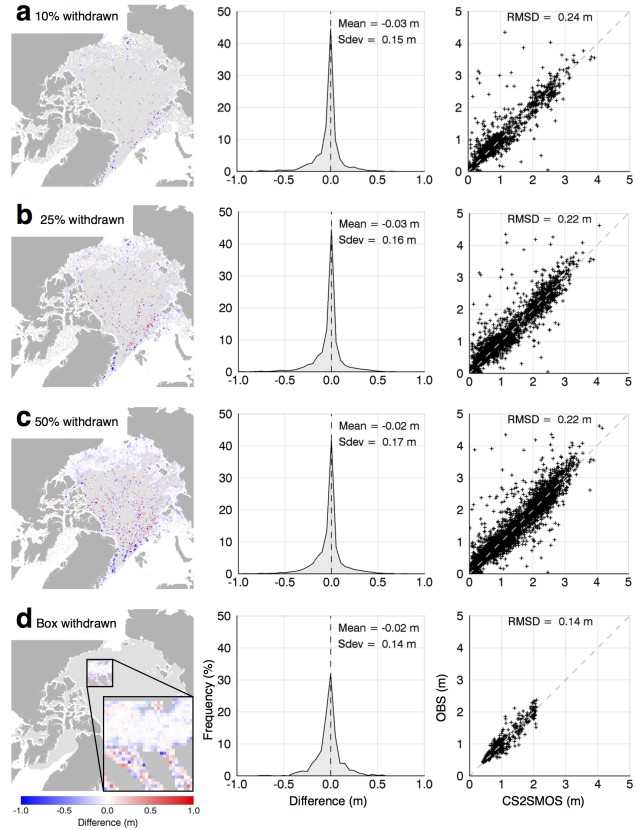

**Figure 12.** Cross-validation experiment, showing the difference between CS2SMOS ice thickness, gridded CryoSat-2 and SMOS observations (OBS) that have been separated in advance as different fractions/areas of withdrawn data: a) 10%, b) 25%, c) 50%, and d) Box. The maps show the withdrawn data subtracted from the CS2SMOS product. The histograms show the differences according to the maps, indicating mean and standard deviation (Sdev) of the differences. Scatter diagrams indicate the root mean square deviation (RMSD).

## 3.2 Cross Validation Experiment

In order to test the robustness of the OI algorithm, we carry out a cross validation. We randomly remove grid cells of observations from the target week (see Figure 5 and 6), with experiments for exclusion of 10% (Figure 12a), 25% (Figure 12b) and 50% (Figure 12c) of both CS2 and SMOS input grid cells. In the fourth case, all data contained in a box in the Western Arctic are withdrawn (Figure 12d). The box intentionally covers both the SMOS and the CS2 domain. After the data omission, the OI algorithm is applied using the reduced target week data set. The maps show the difference between the retrieved CS2SMOS

sea-ice thickness and the withdrawn thickness data for each case. Compared to the SMOS domain, the ice thickness in the CS2 domain in the Central Arctic (Figure 2) reveals a higher level of noise with deviations of up to 1 m. On the other hand, the SMOS domain shows a slightly negative shift of up to 10 cm in some areas. The general pattern remains the same in all cases, independent of the fraction of data that are withdrawn in advance. The shape of the histograms of the differences indicates a



normal distribution with similar standard deviations between 14 and 17 cm. The mean differences are between -2 and -3 cm, which can mostly be attributed to the SMOS domain indicated by the difference maps. The reason is likely the fact that the

5     SMOS background is one week out of phase with the observations (Figure 6), which could cause a small negative bias due to the advancing ice growth. However, in contrast to CS2, the weekly SMOS data coverage during the target week is complete and therefore, this negative bias should not affect the CS2SMOS sea-ice thickness retrieval. The root mean square deviation (rmsd) is 22-24 cm for the first 3 cases and 14 cm for the last case where we separated a box. The smaller rmsd is likely caused by the lack of thicker ice in the chosen box, which does not contain sea ice thicker than about 2 m. This experiment demonstrates the performance of the applied algorithm. In particular, it shows that the background field mostly conserves the mean values even when co-located observations are missing.

## 4    Validation of the merged products with Airborne EM

For validation of WM and CS2SMOS, we use sea-ice thickness measurements obtained during the SMOS-ice 2014 campaign east of the Spitsbergen Archipelago and during the Canadian Arctic Sea Ice Mass Balance Observatory campaign in the Beaufort Sea in April 2016. Surveys have been carried out with an airborne electromagnetic induction thickness sounding

device (EM-Bird) (Pfaffling et al., 2007; Haas et al., 2009; Hendricks, 2009) and are projected and averaged on a 25 km EASE2 grid as given by the satellite products. In addition to the mean AEM thickness in each grid cell, we also calculated the modal AEM thickness. The AEM data set represents total thickness, comprising snow + sea-ice thickness. Therefore, we add the climatological snow depth (W99) to the satellite products. Figure 13 shows the comparison between AEM ice thickness measurements and 4 satellite products at the two validation sites, Beaufort Sea (Figure 13a) and Barents Sea (Figure 13b).

The 4 satellite products are represented by CS2SMOS, WM, SMOS, and CS2. The scatter diagrams illustrate the difference between the satellite products and the corresponding mean and modal AEM thickness. Statistics resulting from Figure 13 are given in Table 3.





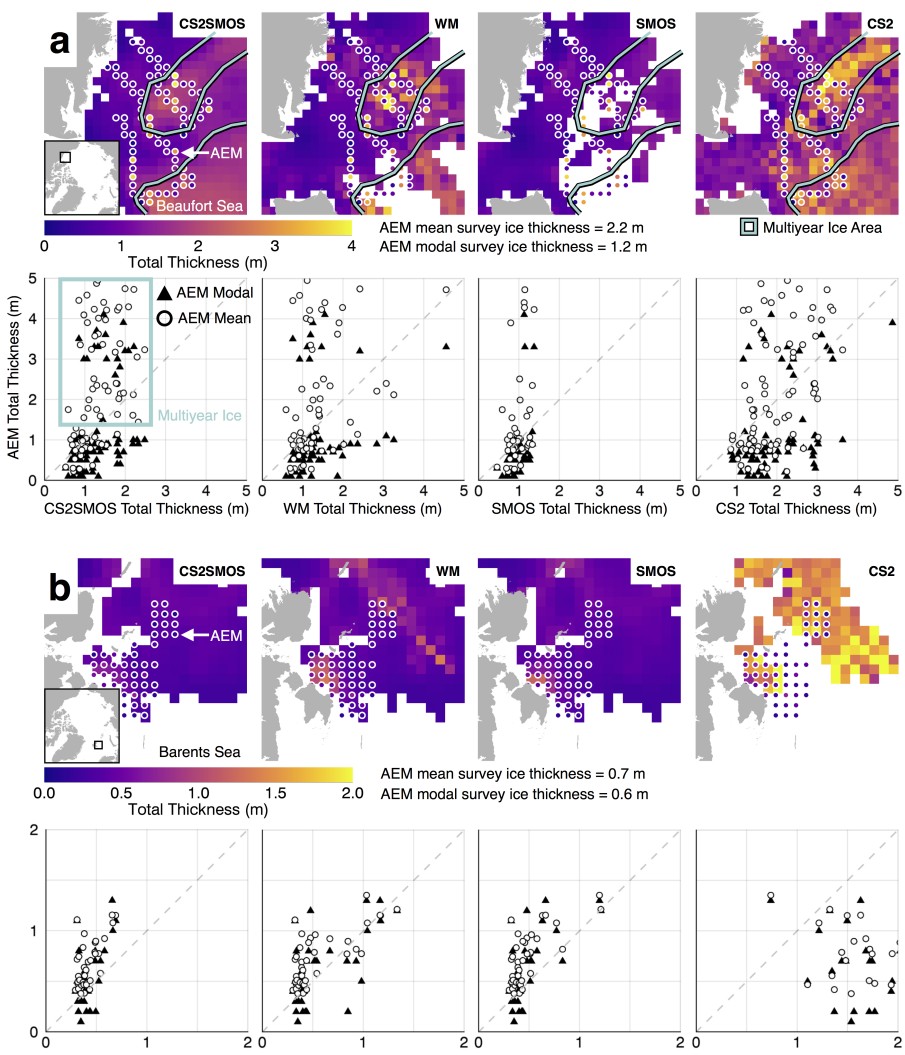

**Figure 13.** Comparison of satellite retrievals with airborne EM thickness measurements (AEM) over a mixed first-year/multiyear ice regime in the Beaufort Sea in April 2016 (a) and over thin ice in the Barents Sea east of Spitsbergen in March 2014 (b). AEM data are compared with optimal interpolation product (CS2SMOS), the weighted mean (WM), the SMOS retrieval, and the monthly CryoSat-2 thickness retrieval (CS2). AEM measurements are averaged on the 25 km EASE2 grid, providing mean and modal total thickness within a grid cell.





**Table 3.** Statistics of the comparison of satellite retrievals with airborne EM thickness measurements (AEM), corresponding to Figure 13. For each case we consider both the AEM modal thickness (AEM mode) and the AEM mean thickness (AEM mean). For the mean bias, AEM measurements are subtracted from the satellite retrievals. rmsd represents the root mean square deviation and r the Pearson correlation coefficient.

| Beaufort Sea | | rmsd (m) | mean bias (m) | r |
|---|---|---|---|---|
| CS2SMOS | AEM mean | 1.56 | -0.86 | 0.48 |
| | AEM mode | 1.03 | 0.11 | 0.36 |
| WM | AEM mean | 1.49 | -0.57 | 0.35 |
| | AEM mode | 1.13 | 0.30 | 0.26 |
| SMOS | AEM mean | 1.16 | -0.38 | 0.37 |
| | AEM mode | 0.75 | 0.19 | 0.46 |
| CS2 | AEM mean | 1.27 | -0.17 | 0.52 |
| | AEM mode | 1.33 | 0.80 | 0.39 |
| Barents Sea | | rmsd | mean bias | r |
| CS2SMOS | AEM mean | 0.30 | -0.25 | 0.65 |
| | AEM mode | 0.26 | -0.11 | 0.60 |
| WM | AEM mean | 0.27 | -0.17 | 0.73 |
| | AEM mode | 0.27 | -0.05 | 0.63 |
| SMOS | AEM mean | 0.30 | -0.24 | 0.7 |
| | AEM mode | 0.27 | -0.11 | 0.67 |
| CS2 | AEM mean | 0.97 | 0.82 | -0.35 |
| | AEM mode | 1.11 | 0.95 | -0.35 |

## 4.1 Beaufort Sea, April 2016

On April 9 and 10, 2 AEM flights were carried out with a fixed wing DC3-T aircraft (Figure 13a). The AEM measurements indicate high mean ice thickness variability ranging between 0.2 m and more than 5 m. Comparing the mean (2.2 m) and modal thickness (1.2 m) of the entire data set indicates substantial deformation. Thickness distribution and OSI SAF ice type data suggest two ice types. First-year ice, reaching a modal thickness of up to 1 m, and multiyear ice with a modal thickness ranging between 2 m and 4 m. The presence of two ice types and the drift along the Beaufort Gyre (Petty et al., 2016) make this region challenging for satellite observations, which are limited in spatial and temporal resolution. Especially scattered thick multiyear ice floes that drift along the Gyre might not be captured by the OSI SAF ice type product, allowing for SMOS thickness estimates in MYI. Therefore, CS2SMOS, WM and SMOS underestimate the mean ice thickness by up to 0.86 m (CS2SMOS). On the other hand, the modal ice thickness is slightly overestimated by up to 0.3 m (WM). It is important to note that WM



and SMOS do not provide a full data coverage. The SMOS data, for example, usually only cover first-year ice. This is also the reason why SMOS exhibits the smallest rmsd for mean and modal thickness (1.16 m and 0.75 m). However, scatter diagrams show good agreement of AEM data and CS2SMOS, WM and SMOS retrievals within the first-year ice, up to about 1.2 m thick ice (Figure 13). CS2 shows the lowest bias (-0.17 m) for the mean ice thickness, but the highest for the modal thickness. The scatter diagrams also indicate that CS2 is not able to capture high thickness gradients due to the presence of scattered heavily deformed multiyear ice, which is transported along with the Beaufort Gyre. As discussed above, the usage of SMOS data in CS2SMOS and WM leads to a stronger underestimation of mean ice thickness of deformed multiyear sea ice, compared to CS2. But it substantially improves the representation of first-year ice thickness. The comparison between WM and CS2SMOS shows that in areas where weekly observations are available, both retrievals show similar agreement with AEM measurements.

## 4.2 Barents Sea, March 2014

Between March 19-26, 8 AEM flights were carried out by a helicopter based on the Norwegian research vessel Lance (King et al., 2016) (Figure 13b). In contrast to the Beaufort Sea data, these data contain first-year ice only. Moreover, the degree of deformation is lower, indicated by only 0.1 m difference between mean and modal thickness of the entire data set. For CS2, the rmsd is 0.97 m for the AEM mean thickness and 1.11 m for the AEM modal thickness, indicating a slightly better representation of the mean thickness in the CS2 product. However, scattering is high and the mean bias of 0.82 m with respect to the mean AEM thickness suggests a strong bias towards thicker ice. Such errors might originate from erroneous sea-surface height interpolation along the CS2 orbits. The SMOS and CS2SMOS retrievals are almost identical for that region, which is caused in part by the better coverage of the SMOS retrieval in that region. In addition, this area is dominated by thin ice, leading to a higher weighting of the SMOS retrieval due to the lower uncertainties (Figure 1). The scatter diagrams reveal a significantly better agreement of the AEM mean thickness measurements with the CS2SMOS, WM and SMOS retrievals (rmsd = 0.27-0.30 m, r=0.65-0.73) than with the CS2 retrieval (rmsd = 0.97, r=-0.35). The observed bias with respect to the mean AEM thickness is -0.25 m for CS2SMOS, -0.17 for WM, and -0.24 m for SMOS, suggesting a bias towards thinner ice. The maps and scatter diagrams indicate that the CS2SMOS, WM and SMOS retrievals capture small thickness gradients visible in the AEM thickness data. This comparison provides evidence that using SMOS data in areas with a thin ice regime will reduce the rmsd and the mean bias when compared to the CS2 product.

## 5 Conclusions

We presented methods to carry out the first joint data merging of CryoSat-2 (CS2) sea-ice thickness fields and thin ice thickness estimates obtained from the L-Band radiometer onboard the Soil Moisture and Ocean Salinity (SMOS) satellite. While CS2 lacks the capability to observe thin ice, SMOS is restricted to ice regimes thinner than about 1 m. We used two approaches for merging CS2 and SMOS ice thickness data: a weighted mean and an optimal interpolation scheme (OI) based on weekly CS2 and SMOS ice thickness grids. While the weighted mean product (WM) only provides estimates at grid cells where observations are available, the OI product (CS2SMOS) provides weekly Arctic-wide sea-ice thickness estimates with corresponding





uncertainty estimates. We have shown that the merged products have the capability to allow for weekly thickness estimates that are sensitive to the entire thickness range, using the complementary sensitivity of the individual products to different thickness regimes. Moreover, the weekly merged products benefit from increased coverage at lower latitudes in conjunction with higher

temporal resolution compared to the CS2 retrieval, which is important for observing ice growth during the freeze-up. In partic-ular, the usage of the combined product will improve thickness retrievals in all areas with thin ice, which we have demonstrated using case studies from the Barents Sea during spring 2014 and Beaufort Sea during spring 2016. Comparisons with airborne electromagnetic thickness measurements (AEM) reveal a reduced root mean square deviation of about 0.7 m for CS2SMOS and WM, compared to the CS2 thickness retrieval in the Barents Sea. Moreover, the comparison shows that retrievals that use

SMOS data seem to capture small thickness gradients in thin ice regimes, whereas the CS2 retrieval is very noisy. In the Bar-ents Sea, the CS2 retrieval overestimates mean thin ice thickness by 0.8 m, while CS2SMOS, WM and SMOS underestimate by about 0.2 m. The comparison with the AEM data has also revealed that WM represents a good estimate in regions where weekly data of SMOS and CS2 are available. For the observation of thicker multiyear ice ($> 1$ m), as in the Beaufort Sea 2016, CS2 provides the best estimates, although limitations in capturing high thickness gradients due to heavily deformed ice exist.

CS2SMOS, however, exclusively provides weekly ice thickness estimates covering the entire Arctic and combining CS2 and SMOS data. The OI approach used in this study can be adopted to merge sea-ice thickness or freeboard data sets derived from other satellite missions, such as the recently launched European Space Agency mission Sentinel-3, which carries a Ku-band radar altimeter similar to SIRAL onboard CS2.

## 6   Data availability

The weekly updated CS2SMOS product, including the weighted means (WM), and the monthly updated CryoSat-2 product are provided at http://www.meereisportal.de. The SMOS ice thickness data are provided at http://icdc.cen.uni-hamburg.de. Sea-ice concentration and Sea-ice type data are provided by OSISAF via http://osisaf.met.no/p/ice/. Barents Sea AEM data are available via doi.org/10.21334/npolar.2016.ee8f4f8d.

*Author contributions.*  Robert Ricker developed the optimal interpolation algorithm and conducted the processing. Stefan Hendricks pro-cessed the CryoSat-2 orbit files. Lars Kaleschke and Xiangshan Tian-Kunze were responsible for the SMOS processing. Jennifer King

5  processed the AEM data in the Barents Sea. Christian Haas processed the AEM data in the Beaufort Sea. Robert Ricker wrote the paper and all Co-authors contributed to the discussion and gave input for writing.

*Competing interests.*  The authors declare no conflict of interest.

*Acknowledgements.*  This work has been conducted in the framework of the European Space Agency project SMOS+ Sea Ice (contracts 4000101476/10/NL/CT and 4000112022/14/I-AM). Lars Kaleschke was responsible for the coordination and design of the SMOS+ Sea Ice



10    project. Many thanks go to Matthias Drusch and the entire SMOS+ Sea Ice Team. Moreover, this study is associated with the Deutsche

Forschungsgemeintschaft (DFG EXC177) and the German Federal Ministry of Economics and Technology (Grant 50EE1008). CryoSat-

2/SMOS data from 2010-2016 are provided by http://www.meereisportal.de (Grant REKLIM-2013-04). J. King was funded by the Norwegian

research council project 'CORESAT' (NFR project number 222681). The helicopter work in the Barents Sea was supported by the Norwegian

Polar Institute (NPI). Thanks also to the crews of RV Lance and Airlift helicopter, and the engineer Marius Bratrein for their help collecting

15    this data.



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
