# Peer review of "A Weekly Arctic Sea-Ice Thickness Data Record from merged CryoSat-2 and SMOS Satellite Data"

_The Cryosphere, 2017_

## Referee Comment (RC1) · Anonymous Referee #1 · 3 Mar 2017

**General comments:**

In this manuscript the authors utilise the complementary sea ice thickness retrieval characteristics of SMOS and CryoSat-2 to produce a weekly merged thickness product. SMOS and CryoSat-2 sea ice thickness results have been published previously, hence this study concentrates on merging the thickness products and evaluating the result. The provision of weekly data, which are optimised for all ice thicknesses, will provide further insight into how different Arctic sea ice classes are changing and potentially benefit modelling studies. The method is, for the most part, rigorous and sound and the authors should be credited for this. However, I have major issues with a.) the structure of the paper and b.) the level of information provided regarding the limitations of each retrieval method and their associated uncertainties. The latter is particularly crucial, as it forms the justification for the body of work presented in the manuscript

The introduction is erratic and contains a wealth of information that would be better placed in Section 2 – Data and Methods. The introduction should be just that – an introduction only – and should summarise the importance of sea ice observations, introduce the SMOS and CyoSat-2 missions and the idea that they are complementary (due to their method and relative uncertainties which will be expanded on later), and introduce the body of the paper. However, uncertainty maps (Figure 1) should not be included before the development of each product and associated error budgets have been described.

Although the authors introduce the CryoSat-2 and SMOS thickness retrievals in Section 2, they do not efficiently and clearly communicate why CryoSat-2 retrievals are reliable over thicker, MYI but not over thinner, FYI, and why the opposite is true for SMOS retrievals. In addition, a full description is needed as to why CryoSat-2 and SMOS uncertainties are complementary – it is not sufficient justification for the work presented to just state that they are/show maps. Some attempt at justification is made in the introduction, but it lacks structure and needs to be significantly expanded. I suggest streamlining the introduction then dividing Section 2.1.1 and 2.1.2 into sub-sections, or something similar. The sub-sections could be arranged as follows:

First sub-section (2.1.1.1 and 2.1.2.1) - retrieval description

\* Outline the retrieval method for CryoSat-2/SMOS

\* Describe theoretical limitations of the sensor for thin/thick ice retrievals, with a clear explanation of the reasons behind the limitations

\* Quantitative stating of thickness limits, or range of (if condition dependent)

Second sub-section (2.1.1.2 and 2.1.1.1) - uncertainty description

- \* Outline the development of the measurement uncertainty budget for CryoSat-2/SMOS
- \* Discuss relative contributions of each input parameter
- \* Explain why this leads to complementary uncertainty compared with the other sensor
\* Currently the authors use statistical uncertainties only for CryoSat-2 to highlight the complementary nature of the two measurement techniques. This is not sufficient. That CryoSat-2 measurements are unreliable over thinner ice, where they are associated with larger uncertainties, forms the major justification of this work. Therefore, the authors need to consider the spatial variations in the actual measurement uncertainty (not just statistical uncertainty) and show that they are larger over thinner ice. Are such uncertainty maps available? If not, can the authors produce them?

Then Figure 1 (measurement error not statistical error) and Figure 2 could be combined, and the reader has all the justification for the work in one place.

Specific comments:

P1 L9: The authors and most readers will know that "narrow-swath altimeter" refers to CryoSat-2 data. However, this is not explained in the manuscript so should not be included here. Just refer to "radar altimeter data" or similar.

P1 L17: "Essential climate variable" is somewhat an ESA tag-phrase and adds nothing to the sentence. Suggest simply changing to "Sea ice affects many climate related processes..."

P2 L3-4: The authors should quantify what is meant by large uncertainties over thin ice regimes, although with improved manuscript structure they might refer straight to uncertainty maps. They reference a paper by Wingham et al (2006), which I don't feel adequately supports their statement. Indeed, the Wingham paper highlights the insensitivity of the ERS altimeters to thin sea ice but the actual discussion of errors only mentions a thickness dependent error when considering observation probability, and states that the magnitude and scale of that error are not easy to estimate before improved resolution measurements from SIRAL.

P2 L13: Define statistical uncertainty, formulaically. This is also relevant for P9 L4 and P10 L8 and is needed to justify the use of gridded data as observations.

TCD
P6 L5: Please briefly justify why SAR and SIN data need to be processed separately, and how

P6 L14: Reference is needed for the theory behind echo separation

P6 L21: What is the modified Warren snow climatology? A clearer explanation is needed of how the authors modify the climatology.

P6 L24: What is meant by the "domain of the W99 climatology"? Despite only being constrained by in situ measurements in the central Arctic, the climatology extends to all Arctic latitudes. This needs to be specifically stated.

P7 L9-10: The final two sentences are far too vague. How is the intensity "almost" independent of incidence angle? This paragraph would benefit from further explanation of the angular dependence of brightness temperature intensity over sea ice.

P8 L4-9: The justification for merging SMOS and Cryosat-2 thicknesses relies on the complementary nature of their uncertainties. Therefore, this paragraph needs expanding. In particular, how can a 100% ice coverage assumption cause underestimation of ice thickness if the condition is not met, and by how much?

P8 L20: How do OSI SAF define ambiguous?

P9 L11: How does OI minimize the total error of observations? This is a key justification for using the OI method to merge the thickness datasets, so needs further explanation.

P10 L10-11: The assumption that ice thicknesses remain static through a week is highly simplified and unlikely. Whilst I appreciate the need to make such assumptions, the authors need to be more transparent about the unlikelihood of this, or provide reference to argue otherwise.

Section 2.3.1: The development of the background field is the key aspect of the method that I am uncertain about. To ensure sufficient coverage, the authors create a background field from CryoSat-2 data extending two weeks before and after the target week.
They do not deem this necessary for SMOS data due to its improved coverage, so only use data from the previous week. What concerns me here is the lack of consistency in the background field time-frames, and the bias it may introduce in the final, interpolated thickness product. Indeed, the authors admit that their cross validation is impacted by the fact that the SMOS background is out of phase with observations (P19 L5) and I'm unconvinced by the authors claim that the negative bias should not affect the CS2SMOS sea ice thickness retrieval. Why do the authors not create temporally complementary CryoSat-2 and SMOS background fields? Have they investigated the impact on the merged product and its evaluation with airborne data of extending the SMOS background field?

Section 3.2: What week/month/year is the cross validation carried out for? Why? The date also needs to be stated in the caption for Figure 12.

P21 L5: It would be interesting to know what fraction of AEM measurements exceed 5 m. It's not possible to tell from Figure 13 as colorbars are capped at 4 m and scatterplots at 5 m, but just stating in the text would be sufficient.

Table 1: Incorrectly states that CS2 (monthly) coverage is Arctic-wide, as due to climatology constraints, measurements are not produced over the Hudson and Baffin Bays. This needs correcting.

Technical comments:

- P1 L1: "Sea-ice thickness on \*\*a\*\* global scale..."
- P2 L16: Should read "Besides the different sensitivities"
- P7 L16: "data as \*\*a\*\* boundary condition
- P9 L8: \*\*modified\*\* climatological snow depth?

P22 L33: "r = 0.65-0.73" and "r = -0.35" i.e. spaces before and after equals sign

---

## Referee Comment (RC2) · Anonymous Referee #2 · 23 Mar 2017

This is an interesting study which merges two sea ice thickness data sets from CryoSat-2 and SMOS since both provide distinct information on sea ice of different thickness ranges. The study is well organized and described. Some of the main points I suggest to address are listed below.

- A detailed description of uncertainties needs to be included since it is a key component to the weighting of the different data sets. In particular the CryoSat-2 uncertainty is not discussed in the manuscript but should be.

- The abstract and conclusion both state that a 0.7 m reduction in RMS deviation in the Barents Sea was observed though it is unclear where this number came from. The data from the Beaufort Sea should be mentioned as well in the abstract if that from the Barents Sea is provided.

[Figure]

- I'm not sure if data exists, but a comparison between the CS2SMOS data and the AEM data would be interesting towards the outer edge of the ice pack where one might expect a different weighting between CS2 and SMOS than in the comparisons done in the central Arctic.

- Some minor grammar mistakes need to be fixed throughout the text.

P2L3-4: CS-2 has been used to retrieve thickness over first year ice as well, the Wingham reference doesn't necessarily support the exclusion of this as well.

Figure 1: How are the uncertainties derived? This needs to be explained in the text.

P6 eqn 3: Is a freeboard correction due to the lower speed of light in snow applied?

P8L2-3: Given the need for cold temperatures, was a mask applied for this or were certain time periods with the data excluded? P10L15: It is stated that the CS-2 data are used from October/November, but what is the starting date used or does it vary? It is not clear in the text.

Eqn. 5: If your observation, analysis, and background fields are all ice thickness you shouldn't need the H operator.

Section 2.3.1: It's not clear to me why you need to construct a spatially continuous background field. If your uncertainties are dependent on distance then you can produce an analysis from the CS-2 and SMOS data at any given point considering the distance between the observations. Presumably this must have been done to fill in the pole hole.

P11L8: It was stated previously that the SMOS algorithm assumes a 100% ice concentration otherwise the measurements are expected to be biased, yet you are using a 15% ice mask in your data. What is the uncertainty introduced by this assumption and how does it impact your retrieval?

Eqns 8-9: I don't understand these equations. What is H in this equation? Equation

9 is defined using matrices on the left-hand side but as a single number on the right hand side, it should be re-written to be mathematically correct.

P14L18: What is the iterative procedure? Optimal interpolation shouldn't involve an iterative method since the solutions are defined equations.

P15L31-32: How is the background field constructed over the pole hole since no data is present there?

Section 3.1: The mean ice thickness values presented, particularly those in Table 2, are for the entire Arctic domain? It would be interesting to see the numbers presented for just the central Arctic Ocean as well since this would give a basis of comparison for other ice thickness data which are typically done for that region.
* * *

---

## Author Comment (AC1) · 15 May 2017

General comments:

In this manuscript the authors utilise the complementary sea ice thickness retrieval characteristics of SMOS and CryoSat-2 to produce a weekly merged thickness product. SMOS and CryoSat-2 sea ice thickness results have been published previously, hence this study concentrates on merging the thickness products and evaluating the result. The provision of weekly data, which are optimised for all ice thicknesses, will provide further insight into how different Arctic sea ice classes are changing and potentially benefit modelling studies. The method is, for the most part, rigorous and sound and the authors should be credited for this. However, I have major issues with a.) the structure of the paper and b.) the level of information provided regarding the limitations of each retrieval method and their associated uncertainties. The latter is particularly crucial, as it forms the justification for the body of work presented in the manuscript

The introduction is erratic and contains a wealth of information that would be better placed in Section 2 – Data and Methods. The introduction should be just that – an introduction only – and should summarise the importance of sea ice observations, introduce the SMOS and CyoSat-2 missions and the idea that they are complementary (due to their method and relative uncertainties which will be expanded on later), and introduce the body of the paper. However, uncertainty maps (Figure 1) should not be included before the development of each product and associated error budgets have been described.

Although the authors introduce the CryoSat-2 and SMOS thickness retrievals in Section 2, they do not efficiently and clearly communicate why CryoSat-2 retrievals are reliable over thicker, MYI but not over thinner, FYI, and why the opposite is true for SMOS retrievals. In addition, a full description is needed as to why CryoSat-2 and SMOS uncertainties are complementary – it is not sufficient justification for the work presented to just state that they are/show maps. Some attempt at justification is made in the introduction, but it lacks structure and needs to be significantly expanded. I suggest streamlining the introduction then dividing Section 2.1.1 and 2.1.2 into sub-sections, or something similar. The sub-sections could be arranged as follows:

First sub-section (2.1.1.1 and 2.1.2.1) – retrieval description

* Outline the retrieval method for CryoSat-2/SMOS

* Describe theoretical limitations of the sensor for thin/thick ice retrievals, with a clear explanation of the reasons behind the limitations

* Quantitative stating of thickness limits, or range of (if condition dependent)

Second sub-section (2.1.1.2 and 2.1.1.1) – uncertainty description

* Outline the development of the measurement uncertainty budget for CryoSat-2/SMOS

* Discuss relative contributions of each input parameter

* Explain why this leads to complementary uncertainty compared with the other sensor

\* Currently the authors use statistical uncertainties only for CryoSat-2 to highlight the complementary nature of the two measurement techniques. This is not sufficient. That CryoSat-2 measurements are unreliable over thinner ice, where they are associated with larger uncertainties, forms the major justification of this work. Therefore, the authors need to consider the spatial variations in the actual measurement uncertainty (not just statistical uncertainty) and show that they are larger over thinner ice. Are such uncertainty maps available? If not, can the authors produce them?

Then Figure 1 (measurement error not statistical error) and Figure 2 could be combined, and the reader has all the justification for the work in one place.

We thank the reviewer for these thoughtful and constructive comments. We tried to improve the structure of the paper in order to meet your suggestions. We have streamlined the introduction and shifted the part about the product uncertainties and complementarity to section 2. Now, product uncertainties and complementarity are only described briefly in the introduction, and then in more detail in section 2, where we inserted a new subsection: „2.1.3 Complementarity of CryoSat-2 and SMOS Sea-Ice Thickness Products". We have chosen this way in order to avoid introducing another subsection level (2.1.1.2 and 2.1.1.1).

Regarding the comment about the statistical uncertainties: we agree that the term "statistical" is misleading here, since it might also mean an uncertainty in the sense of standard deviation of observations or similar. Therefore, we decided to use the term "observational" uncertainty instead, throughout the manuscript. Indeed, CryoSat-2 uncertainties are derived by quantifying different sources of uncertainty in each single measurement, as speckle noise, uncertainty in sea-surface height determination, ice and snow densities, etc.. An observational thickness uncertainty is then found by gaussian error propagation. More details can be found in Ricker et al. (2014). We added a paragraph for further explanation. Therefore, to be more specific, the maps in Figure 1 (original version) are already the requested CryoSat-2 actual measurement uncertainty maps. There are different reasons why uncertainty over thin ice is higher in the CryoSat-2 product:

1. Thinner ice rather occurs in lower latitudes where, due to the CryoSat-2 orbit inclination, the density of measurements is lower than closer to the pole where ice is thicker. This is important as, by gridding, measurement uncertainties are reduced by spatial averaging and the uncertainty reduction depends on the number of available measurements within a grid cell.

2. The relative uncertainty increases over thin ice, as measurement uncertainties do not decrease over thinner ice.

3. In the marginal ice zones, when ice concentration decreases, many openings in the sea ice cover can lead to an underrepresentation of (thin) sea ice.

4. With many openings in the sea ice (as in the marginal ice zones), so called "snagging" leads to increased uncertainties in the range measurements (Armitage and Davidson, 2014)

In the following we briefly list major changes in the document:

- We shifted the discussion about the complementarity of both thickness products from the Introduction to Data and Methods and inserted a new subsection there: **2.1.3 Complementarity of CryoSat-2 and SMOS Sea-Ice Thickness Products**. Therefore, also the order of Figure 1 and Figure 2 has switched.

- Figure 1 has been updated, we accidentally plotted the SMOS relative uncertainty of March also for November. This has been fixed.

- For the background field, we now also included SMOS retrievals from the week after the target week in order to avoid a potential bias. As a consequence, also Figures 5 and 6 have been updated. However, the effect is minimal as shown by the changes of statistics in Tables 2 and 3.

- Figure 12 has been updated, since we found a bug in the plotting routine. Therefore, values have slightly changed.

NOTE: All tracked changes in the manuscript are attached to this response letter.

Specific comments:

P1 L9: The authors and most readers will know that "narrow-swath altimeter" refers to CryoSat-2 data. However, this is not explained in the manuscript so should not be included here. Just refer to "radar altimeter data" or similar.

We deleted "narrow-swath" as suggested.

P1 L17: "Essential climate variable" is somewhat an ESA tag-phrase and adds nothing to the sentence. Suggest simply changing to "Sea ice affects many climate related processes. . ."

We removed "Essential climate variable" as suggested.

P2 L3-4: The authors should quantify what is meant by large uncertainties over thin ice regimes, although with improved manuscript structure they might refer straight to uncertainty maps. They reference a paper by Wingham et al (2006), which I don't feel adequately supports their statement. Indeed, the Wingham paper highlights the insensitivity of the ERS altimeters to thin sea ice but the actual discussion of errors only mentions a thickness dependent error when considering observation probability, and states that the magnitude and scale of that error are not easy to estimate before improved resolution measurements from SIRAL.

We added a quantification to this statement. Referring to Figure 1 (original version), the relative uncertainties over thin ice (< 1 m) are in the range of 100 % or above. We also added the reference Ricker et al. (2014), since it explicitly discusses uncertainties in the CS2 ice thickness product.

P2 L13: Define statistical uncertainty, formulaically. This is also relevant for P9 L4 and P10 L8 and is needed to justify the use of gridded data as observations.

As above, we agree that the term "statistical uncertainty" is somewhat confusing and replaced it with "observational uncertainty". We added more explanation and briefly explain the contributions to the "observational uncertainty".

P6 L5: Please briefly justify why SAR and SIN data need to be processed separately, and how.

We added a short explanation in the manuscript: "…but must be processed separately, as we discard the phase information of SIN waveforms (Kurtz et al. (2014)."

P6 L14: Reference is needed for the theory behind echo separation.

We included a reference to Laxon et al. (2003).

P6 L21: What is the modified Warren snow climatology? A clearer explanation is needed of how the authors modify the climatology.

With "modification", we mean the reduction of snow depth over first-year ice by a factor of 50 %, suggested by Kurtz and Farrell (2011). We changed this sentence for clarification.

P6 L24: What is meant by the "domain of the W99 climatology"? Despite only being constrained by in situ measurements in the central Arctic, the climatology extends to all Arctic latitudes. This needs to be specifically stated.

The W99 climatology is a 2d polynomial fit based on the in situ observations, which is applicable to all latitudes. But the 2d fit will be just extrapolated in regions where no observations are available. Therefore, values in these regions are not reliable. With "domain", we therefore refer to the areas where observations can be used as tie points for the 2d fit. We added a short explanation in the manuscript.

[Figure]

Fig. 1. Vertically (V) and horizontally (H) polarized Tbs and the intensity as a function of incidence angle, calculated using a three-layer model for sea ice with $d_{ice}$ = 1 m, a bulk salinity of $S_{ice}$ = 8 gkg$^{-1}$, and a bulk ice temperature of $T_{ice}$ = -7 °C.

P7 L9-10: The final two sentences are far too vague. How is the intensity "almost" independent of incidence angle? This paragraph would benefit from further explanation of the angular dependence of brightness temperature intensity over sea ice.

As can be seen in Figure 1 in this document (which is Fig. 3 in Tian-Kunze et al., 2014), the vertically polarized brightness temperature increases with increasing incidence angle, whereas horizontally polarized brightness temperature decreases. This counteract behavior results in that the intensity, which is the average of both polarizations, remains almost constant in the incidence angle range of 0-40 degree.

P8 L4-9: The justification for merging SMOS and Cryosat-2 thicknesses relies on the complementary nature of their uncertainties. Therefore, this paragraph needs expanding.

We added following text to the uncertainty estimation part in the revised version:

„The SMOS uncertainty given in the v3 product is estimated based on the uncertainty in the input parameters in the thermodynamic and radiation model as well as in the thickness distribution function. At present, the estimation was carried out for each parameter - brightness temperature, ice temperature and ice salinity respectively, by keeping the other parameters constant. The uncertainty given in the product is then the sum of uncertainties caused by each parameter. In v3, we also varied the sigma in the lognormal ice thickness distribution function, which is used to convert plane layer ice thickness into heterogenous layer mean ice thickness in the retrieval. This uncertainty is then added to the overall uncertainties caused by the brightness temperature, ice temperature and ice salinity. Errors caused by the assumptions about fluxes and snow thickness have not yet been included."

In particular, how can a 100% ice coverage assumption cause underestimation of ice thickness if the condition is not met, and by how much?

This is discussed in detail in Tian-Kunze et al., 2014 (see section 4.1). Brightness temperature over ice-sea water mixed areas can be described as

$$TB = TB_{water} \times (1-IC) + TB_{ice} \times IC, \quad (1)$$

where IC is the ice concentration, $TB_{water}$ and $TB_{ice}$ are the brightness temperatures over sea water and sea ice respectively. $TB_{ice}$ is about 100 K higher than $TB_{water}$, which means that under 100 % ice coverage assumption, we underestimate the $TB_{ice}$, leading to lower ice thickness. In Tian-Kunze et al., 2014, we showed that by using eq. 1 and the radiation model, the underestimation of ice thickness increases exponentially with decreasing ice concentration. However, a comparison with MODIS-derived ice thickness in the Kara Sea has shown that observational uncertainty of SMOS ice thickness under low ice concentration condition is as low as 10 cm (see SMOS + Sea Ice final report). A more deliberate estimation of SMOS ice thickness bias and uncertainty remains as future work.

The 100% ice coverage assumption is used because the accuracy of the ice concentration product is limited and by using the product ice concentration, one will introduce larger errors than with the 100 % assumption. Moreover, this issue only accounts for a minor area, while during winter, most of the ice covered area in the Arctic has ice concentrations (IC) higher than 90 % (Andersen et al., 2007).

P8 L20: How do OSI SAF define ambiguous?

The OSI SAF ice type is defined "ambiguous" when their Multi sensor ice type analysis approach has problems to differ between FYI and MYI. This is mostly the case in the transition zone between FYI and MYI regime where the ice type is mixed. For more details, we refer to the OSI SAF product manual:

http://osisaf.met.no/docs/osisaf_cdop2_ss2_pum_sea-ice-edge-type.pdf

P9 L11: How does OI minimize the total error of observations? This is a key justification for using the OI method to merge the thickness datasets, so needs further explanation.

The OI provides an estimate of sea ice thickness at a given location using a linear combination of arbitrarily distributed thickness observations. These observations are weighted so that the expected uncertainty of the thickness estimate is a minimum in the least squares sense. With Eq. 8, we calculate the weights K that are needed to minimize error variances. We added a sentence for clarification.

P10 L10-11: The assumption that ice thicknesses remain static through a week is highly simplified and unlikely. Whilst I appreciate the need to make such assumptions, the authors need to be more transparent about the unlikelihood of this, or provide reference to argue otherwise.

For many areas in the Arctic, ice motion during one week will be in the range of one or two grid cells (25-50 km), except Fram Strait and parts of the Beaufort Gyre. Considering the uncertainties of the satellite retrievals, we conclude that this assumption is acceptable, though including ice motion data in the optimal interpolation might still lead to improvements and should be investigated in future.

Section 2.3.1: The development of the background field is the key aspect of the method that I am uncertain about. To ensure sufficient coverage, the authors create a background field from CryoSat-2 data extending two weeks before and after the target week.

They do not deem this necessary for SMOS data due to its improved coverage, so only use data from the previous week. What concerns me here is the lack of consistency in the background field time-frames, and the bias it may introduce in the final, interpolated thickness product. Indeed, the authors admit that their cross validation is impacted by the fact that the SMOS background is out of phase with observations (P19 L5) and I'm unconvinced by the authors claim that the negative bias should not affect the CS2SMOS sea ice thickness retrieval. Why do the authors not create temporally complementary CryoSat-2 and SMOS background fields? Have they investigated the impact on the merged product and its evaluation with airborne data of extending the SMOS background field?

Thank you for pointing on this issue. We are aware of this and now also included the SMOS retrieval from the week after the target week in order to generate the background field (see modified Figure 6). However, the effect is very small (in the range of few cm). We repeated the cross validation and still receive the negative bias. Moreover, we found a bug in the plotting routine of Figure 12. We fixed this, but the main results are the same. Reconsidering the negative bias in the cross validation, we found the following explanation: The CS2 and SMOS retrieval domains are not symmetric due to the fact that we have to truncate the SMOS retrieval for thick ice, since the methodical approach does not apply for thick ice. On the other hand, the CS2 retrieval is used over the entire thickness range, but with higher uncertainties over thin ice. Therefore, CS2 thickness over thin ice is mostly reduced by the SMOS retrieval, while in contrast this is barely the case for SMOS data over thick ice, because we truncate the retrieval. Hence, due to the optimal interpolation, there will be always a negative bias when doing the cross validation experiment with the original input data from CS2 and SMOS. We included an explanation to the manuscript.

Section 3.2: What week/month/year is the cross validation carried out for? Why? The date also needs to be stated in the caption for Figure 12.

We thank the reviewer for pointing on this. We added the date to the caption.

P21 L5: It would be interesting to know what fraction of AEM measurements exceed 5 m. It's not possible to tell from Figure 13 as colorbars are capped at 4 m and scatter- plots at 5 m, but just stating in the text would be sufficient.

We capped the ranges in the figures in order to improve the visibility of the differences between the AEM data and the satellite derived products. Indeed, one AEM (gridded) measurement point of 5.6 m ice thickness does not appear in the scatter plot. We added a note in the caption of Figure 13 that the reader is aware.

Table 1: Incorrectly states that CS2 (monthly) coverage is Arctic-wide, as due to climatology constraints, measurements are not produced over the Hudson and Baffin Bays. This needs correcting.

Thanks for pointing on this. Although, CS2 measurements over these areas exist and by using other climatologies or even snow depth products, sea-ice thickness might be available with decent uncertainties in future. However, we added a comment in the "Notes and applicability" -column: "constraints in regions where snow climatologies are unavailable".

Technical comments:
P1 L1: "Sea-ice thickness on **a** global scale. . ."
P2 L16: Should read "Besides the different sensitivities"
P7 L16: "data as **a** boundary condition
P9 L8: **modified** climatological snow depth?
P22 L33: "r = 0.65-0.73" and "r = -0.35" i.e. spaces before and after equals sign

We included the list of technical comments. Thank you.

[revised manuscript text omitted]

---

## Author Comment (AC2) · 15 May 2017

This is an interesting study which merges two sea ice thickness data sets from CryoSat- 2 and SMOS since both provide distinct information on sea ice of different thickness ranges. The study is well organized and described. Some of the main points I suggest to address are listed below.

- A detailed description of uncertainties needs to be included since it is a key component to the weighting of the different data sets. In particular the CryoSat-2 uncertainty is not discussed in the manuscript but should be.

We thank the reviewer for the thoughtful comments, which significantly helped to improve the manuscript. We agree that the information about the CryoSat-2 uncertainties are sparse and since they are crucial for the methodical approach, we added more information in section 2.

In the following we briefly list major changes in the document:

- We shifted the discussion about the complementarity of both thickness products from the Introduction to Data and Methods and inserted a new subsection there: 2.1.3 Complementarity of CryoSat-2 and SMOS Sea-Ice Thickness Products. Therefore, also the order of Figure 1 and Figure 2 has switched.
- Figure 1 has been updated, we accidentally plotted the SMOS relative uncertainty of March also for November. This has been fixed.
- For the background field, we now also included SMOS retrievals from the week after the target week in order to avoid a potential bias. As a consequence, also Figures 5 and 6 have been updated. However, the effect is minimal as shown by the changes of statistics in Tables 2 and 3.
- Figure 12 has been updated, since we found a bug in the plotting routine. Therefore, values have slightly changed.

NOTE: All tracked changes in the manuscript are attached to this response letter.

- The abstract and conclusion both state that a 0.7 m reduction in RMS deviation in the Barents Sea was observed though it is unclear where this number came from. The data from the Beaufort Sea should be mentioned as well in the abstract if that from the Barents Sea is provided.

This number refers to Table 3, where the difference between rmsd (AEM-CS2SMOS) and rmsd (AEM-CS2) is about 0.7 m for the mean ice thickness in the Barents Sea. It is true that the 0.7 m reduction in rmsd ist not stated explicitly in the test and therefore we added a sentence. Moreover, we now also refer to the results in the Beaufort Sea in the conclusion and the abstract.

- I'm not sure if data exists, but a comparison between the CS2SMOS data and the AEM data would be interesting towards the outer edge of the ice pack where one might expect a

different weighting between CS2 and SMOS than in the comparisons done in the central Arctic.

Unfortunately, AEM measurements mainly exist for the central Arctic. However, for this reason we have chosen Barents Sea and Beaufort Sea as validation sites. The Barents Sea is dominated by the SMOS ice thickness product, while in the Beaufort Sea, we find scattered multiyear ice floes (up to 5 m thickness) within an area of about 1 m thick first-year ice. Such an area with mixed ice types and high drift rates is very challenging but a perfect test case for the data merging, since both data products contribute significantly.

- Some minor grammar mistakes need to be fixed throughout the text.

Thank you for pointing on this. We tried to improve the grammar throughout the manuscript.

P2L3-4: CS-2 has been used to retrieve thickness over first year ice as well, the Wingham reference doesn't necessarily support the exclusion of this as well.

We added a sentence here and now also state that CS2 can be used for retrieving firstyear ice thickness, though the uncertainties over thin ice regimes and in the marginal ice zones are considerably high.

Figure 1: How are the uncertainties derived? This needs to be explained in the text.

We added further explanation in section 2. Moreover, we followed the suggestions of reviewer #1 and moved the discussion about uncertainties and product complementarity to section 2.

P6 eqn 3: Is a freeboard correction due to the lower speed of light in snow applied?

Yes, such a correction is applied. We added a short note in the text.

P8L2-3: Given the need for cold temperatures, was a mask applied for this or were certain time periods with the data excluded? P10L15: It is stated that the CS-2 data are used from October/November, but what is the starting date used or does it vary? It is not clear in the text.

The SMOS data product is available from mid October on (October 15). CS-2 data are processed from October on as well. Since we use calendar weeks, the starting date of the merged product will vary but corresponds with the second week of SMOS thickness retrievals (second half of October), since the first week of SMOS thickness retrievals is used for the background field. We corrected the text and now state that the starting date is end of October.

Eqn. 5: If your observation, analysis, and background fields are all ice thickness you shouldn't need the H operator.

The H Operator here only projects (interpolates) the grid cells of the background field thickness onto the locations of the observations, which can be different. For example, the SMOS (observations) are provided on a 12.5 km polarstereographic grid. Finally, H formally just ensures that for each point of the observational vector, we have exactly one (interpolated) point from the background field.

Section 2.3.1: It's not clear to me why you need to construct a spatially continuous background field. If your uncertainties are dependent on distance then you can produce an analysis from the CS-2 and SMOS data at any given point considering the distance between the observations. Presumably this must have been done to fill in the pole hole.

The background field is needed as a "first guess" (see equation 5). For each observational data point, a background value is needed, the pole hole of the background field is filled by an inverse-distance interpolation. In other words, the observations of the target week are used to "update" the background field. The background field is also important for the gaps between orbits in the weekly CS-2 product. Without the background field, they would be just filled by linear interpolation, causing significant biases.

P11L8: It was stated previously that the SMOS algorithm assumes a 100% ice con- centration otherwise the measurements are expected to be biased, yet you are using a 15% ice mask in your data. What is the uncertainty introduced by this assumption and how does it impact your retrieval?

This assumption leads to a underestimation of ice thickness in the merged product when ice concentration is low. It is our goal to correct this effect in a future revision of the product, but before, the relationship between low ice concentration and underestimation of SMOS ice thickness needs to be investigated in more detail. Roughly, assuming 10 cm SMOS ice thickness when ice concentration is about 70 %, the true ice thickness is rather 14 cm. However, the uncertainties in the ice concentration products make it difficult to correct for this accordingly.

Eqns 8-9: I don't understand these equations. What is H in this equation? Equation 9 is defined using matrices on the left-hand side but as a single number on the right hand side, it should be re-written to be mathematically correct.

Thank you for pointing on this. Indeed, on the right-hand side we refer to the elements of the matrices on the left-hand side. We corrected this.

P14L18: What is the iterative procedure? Optimal interpolation shouldn't involve an iterative method since the solutions are defined equations.

Thank you for reviewing the equations. Basically, the OI can be accomplished just using matrices but the they would become too large, which would prevent or significantly slowing down computation on a common machine. However, the formal notation here is wrong and we corrected this. The iterative procedure involves the iterative calculation of each analysis grid point, instead of computing it in one step.

P15L31-32: How is the background field constructed over the pole hole since no data is present there?

Here we use an inverse distance interpolation with a weighting power of the distance of 3.

Section 3.1: The mean ice thickness values presented, particularly those in Table 2, are for the entire Arctic domain? It would be interesting to see the numbers presented for just the central Arctic Ocean as well since this would give a basis of comparison for other ice thickness data which are typically done for that region.

Yes, these values are for the entire Arctic domain. We agree that investigating the thickness distributions of different regions would be interesting. This could be done in follow-on studies. In this study, we want to focus on methodical approach. Moreover, we provide these data on a public server, mean values for the central Arctic or any other regions can be calculated by the user.

**A Weekly Arctic Sea-Ice Thickness Data Record from merged CryoSat-2 and SMOS Satellite Data**

Robert Ricker1,2, Stefan Hendricks1, Lars Kaleschke3, Xiangshan Tian-Kunze3, Jennifer King4, and Christian Haas1,5

1Alfred Wegener Institute, Helmholtz Centre for Polar and Marine Research, Bremerhaven, Bussestrasse 24, 27570
 Bremerhaven, Germany
 2Univ. Brest, CNRS, IRD, Ifremer, Laboratoire d'Oceanographie Physique et Spatiale (LOPS), IUEM, 29280, Brest, France
 3Institute of Oceanography, University of Hamburg, Bundesstrasse 53, 20146 Hamburg, Germany
 4Norwegian Polar Institute, Tromsoe, Norway
 5York University, Toronto, Canada

Correspondence to: Robert Ricker (Robert.Ricker@awi.de)

**Abstract.** Sea-ice thickness on a global scale is derived from different satellite sensors using independent retrieval methods. Due to the sensor and orbit characteristics, such satellite retrievals differ in spatial and temporal resolution as well as in the sensitivity to certain sea-ice types and thickness ranges. Satellite altimeters, such as CryoSat-2 (CS2), sense the height of the ice surface above the sea level, which can be converted into sea-ice thickness. However, relative Relative uncertainties associated

- 5 with this method are large over thin ice regimes. Another retrieval strategy is realized by method is based on the evaluation of surface brightness temperature in L-Band microwave frequencies (1.4 GHz) with a thickness-dependent emission model, as measured by the Soil Moisture and Ocean Salinity (SMOS) satellite. While the radiometer based method looses sensitivity for thick sea ice (> 1m), relative uncertainties over thin ice are significantly smaller than for the altimetry-based retrievals. In addition, the SMOS product provides global sea-ice coverage on a daily basis unlike the narrow-swath-altimeter data. This
- 10 study presents the first merged product of complementary weekly Arctic sea-ice thickness data records from the CS2 altimeter and SMOS radiometer. We use two merging approaches: a weighted mean and an optimal interpolation scheme (OI). While the weighted mean leaves gaps between CS2 orbits, OI is used to produce weekly Arctic-wide sea-ice thickness fields. The benefit of the data merging is shown by a comparison with airborne electromagnetic induction sounding measurements. When compared to airborne thickness data in the Barents Sea, the merged products reveal a reduced product has a root mean square
- 15 deviation (rmsd) of about 0.7 m compared to less than the CS2 retrieval product 
[revised manuscript text omitted]